# The Effect of Cooling Rate on Crystallographic Features of Phase Transformations in Zr-2.5Nb

**DOI:** 10.3390/ma16103758

**Published:** 2023-05-16

**Authors:** Mikhail L. Lobanov, Valentin Yu. Yarkov, Vladimir I. Pastukhov, Inna A. Naschetnikova, Stepan I. Stepanov, Andrey A. Redikultsev, Mariya A. Zorina

**Affiliations:** 1Heat Treatment & Physics of Metals Department, Ural Federal University, 19 Mira St., 620062 Ekaterinburg, Russia; valick99@gmail.com (V.Y.Y.);; 2Institute of Nuclear Materials, 624250 Zarechny, Russia

**Keywords:** zirconium, burgers relationship, twin, misorientation analysis, hexagonal close-packed structure

## Abstract

Zirconium (Zr) alloys are utilized as structural components for the cores of nuclear reactors due to the excellent combination of their mechanical properties and corrosion resistance under intense neutron irradiation conditions in water. The characteristics of microstructures formed during heat treatments play a crucial role in obtaining the operational performance of parts made from Zr alloys. This study investigates the morphological features of (α + β)-microstructures in the Zr-2.5Nb alloy, as well as the crystallographic relationships between α- and β-phases. These relationships are induced by the β→α(α″) displacive transformation that occurs during water quenching (WQ) and the diffusion-eutectoid transformation that takes place during furnace cooling (FC). To conduct this analysis, samples solution treated at 920 °C were examined using EBSD and TEM. The experimental distribution of α/β-misorientations for both cooling regimes deviates from the Burgers orientation relationship (BOR) at a discrete set of angles close to 0, 29, 35, and 43°. The experimental α/β-misorientation spectra are confirmed by crystallographic calculations for the β→α→β-transformation path based on the BOR. Similar spectra of misorientation angle distribution in α-phase and between α and β phases in Zr-2.5Nb after WQ and FC point to similar transformation mechanisms and the significant role of shear and shuffle in β→α-transformation.

## 1. Introduction

Since the 1950s, Zr-based alloys have been extensively employed in structures operating in the aggressive environments of nuclear reactor cores and critical chemical industry products, owing to their unique combination of mechanical properties, corrosion resistance, and low neutron absorption rates [1,2,3,4,5,6,7]. Currently, Zr alloy products serve as benchmarks for the functional properties of new materials in the nuclear industry [8,9]. Their potential bolsters interest for further medical applications in Zr-based alloys [10,11,12].

Nuclear reactor technologies are progressing towards fuel conservation, reduced fuel cycle costs, and enhanced reactor safety and reliability. Consequently, higher performance characteristics are demanded for Zr alloys used as fuel rod cladding materials in radiation zones [3,13,14,15]. Ongoing research into the development of Zr alloys and optimization of their microstructure addresses several key challenges: controlling radiation growth [4,16], increasing corrosion resistance [2,3,6,7], and reducing hydrogen absorption [2,3]. Radiation growth refers to the dimensional change of Zr alloy reactor components, even in the absence of external stresses influenced by the size, shape of grains, and crystallographic texture of the product obtained during manufacturing [4]. Note that the presence of Nb in the solid α-solution reduces radiation growth [4].

As demonstrated in [6,7,17,18,19], the corrosion resistance of Zr alloy products, primarily hydrogenation and oxidation rates, strongly depends on the product microstructure and texture. In particular, [3] microstructural states with the presence of β-Zr in a nuclear Zr component are generally undesirable from a corrosion resistance standpoint.

Hydride precipitation studies in Zr alloys encompassing first-principles calculations [20,21] have been conducted since the 1960s and continue today. Due to the low solubility of hydrogen in the crystal lattice of α-Zr (0.034 ppm at 293 K), according to [2], hydrogen penetration into the product results in hydride precipitation, potentially causing significant embrittlement and subsequent failure of the components in a nuclear reactor [2,3,20]. The shape and orientation of hydride precipitates affect the product impact toughness [20,21,22,23]. Moreover, hydrides can be reoriented under changing external stresses during processing, which may alter the anisotropy of the mechanical properties of the final product [24].

Manufacturing zirconium alloy components for nuclear reactors involves numerous thermomechanical treatments, with the resulting microstructure determining the product’s long-term and short-term properties [13]. The microstructure of Zr alloys, including local and integral texture states, crucially affects functional properties, prompting research into obtaining various microstructures through heat treatments, primarily by altering cooling rates from the β-region [14,25,26].

In [14], the influence of the microstructure on the mechanical properties is analyzed for the Zr-4Hf-3Nb alloy. It was shown how an increase in the cooling rate changes the mechanism of phase transformation (β→α, ω, α′) due to the refinement of the microstructure and reduction of the fraction of β-phase precipitates with a change in its chemical composition.

In [25], the study of the Zr-2.5Nb alloy demonstrated that despite variations in the morphology and size of the formed α(α″)-phases at different cooling rates, the transformation of the crystal lattice proceeds according to the Burgers orientation relationship (BOR). In this context, rapid cooling results in the realization of all 12 possible variants of the hexagonal close-packed (HCP) α-phase predetermined by the orientation of the parent body-centered cubic (BCC) β-phase grain.

In [26], the microstructures of the novel Zr-4Sn-4Nb and Zr-8Sn-4Nb alloys are highlighted for their similarity to the microstructures of (α + β) titanium alloys. The potential for tailoring microstructures through heat treatment processes is also demonstrated.

In addition, Zr-Nb alloys serve as an exceptional subject for fundamental research, owing to the classical variety of equilibrium transformations present in the Zr-Nb system (congruent, polymorphic, monotectoid, eutectoid) [27]. Another aspect of Zr-Nb alloy research pertains to the potential for obtaining metastable phases (α″, β-Zr, β-Nb, ω) [27,28], as well as nanocrystalline and amorphous states via heat treatment [29,30,31]. The formation of α(α″) is contingent upon the Nb content and cooling rate. It can occur through either the diffusionless martensitic mechanism or the “bainite” mechanism, which entails a diffusion redistribution of alloying elements.

The martensitic transformation occurs at the speed of sound propagation within the material and results in the formation of a metastable α″-phase [27,28]. Physically, the martensitic transformation of the high-temperature β-phase is driven by the short-wavelength instability associated with the transverse phonon ½[110] with the wave vector ½<110> and the polarization vector parallel to <1$\overset{-}{1}$0>, which induces the shuffling of {110} planes in the BCC-lattice according to the Burgers mechanism [32,33]. Concurrently, atomic displacements are related to an acoustic transverse phonon with a vector parallel to <112> and a polarization vector parallel to <11$\overset{-}{1}$>. A shift in the {110} planes in opposite directions <1$\overset{-}{1}$0> at a distance of a·(2)^1/2^/12 leads to the occurrence of the BCC→HCP transformation. Two equivalent long-wavelength shifts ((11$\overset{-}{2}$)[$\overset{-}{1}$11] and ($\overset{-}{1}\overset{-}{1}$2)[1$\overset{-}{1}$1]) enable a change in the angle between [$\overset{-}{1}$11] and [$\overset{-}{1}\overset{-}{1}$1] in the β-phase from 70.5° to 60° between [$\overset{-}{2}$110] and [$\overset{-}{1}\overset{-}{1}$20] in the α-phase in the equivalent {110} β||{0001}α planes required for the β→α transformation [34]. It is crucial to note that, regardless of the transformation mechanism (martensitic or diffusion-controlled), the BOR ({110}β||{0001}α, <111>β||<11$\overset{-}{2}$0>α) and the habit plane—{334} are essentially fulfilled [27,28,35].

The implementation of the multivariant β→α″(α)-transformation (1 orientation of the β-phase translates into 12 orientations of the α″(α)-phase) in accordance with the BOR does not imply the formation of a crystallographic texture. However, the formation of a crystallographic texture during standard supertransus solution treatment of the textured material as a result of α→β-transformation upon heating (1α→6β) and β→α″(α) upon cooling (6β→6 × 12 = 48α) is a well-known fact, with a potential mechanism illustrated in [36]. Additionally, the structural-textural inheritance during multiple phase transformations was demonstrated at the Ti-6Al-4V alloy in [37]. In [38,39,40], similar texture inheritance in steels and bronzes is explained by the formation of new phase nuclei on crystallographically ordered boundaries near the coincidence site lattice (CSL) boundaries (e.g., ∑3, ∑33, etc.), which arise in the material as a result of prior deformation (between stable deformation orientations) or due to phase transformations realized in accordance with the OR.

Tailoring the crystallographic patterns of local texture states in a material, along with characteristics of the phase composition, grain shape, and size during phase transformations induced by heat treatments, allow for designing material or product microstructures for various applications. This work studies the crystallographic features of phase transformation and morphological aspects of microstructure formation in the Zr-2.5Nb alloy during cooling from a single-phase β-region with different rates.

## 2. Materials and Methods

Rectangular samples with dimensions of 3.3 × 5.0 × 4.0 mm were separated from the commercial hot deformed and air-cooled Zr-2.5Nb pipe using the electrical discharge machine Ecocut. The β-transus of 900 ± 10 °C and eutectoid transformation temperature of 680 ± 10 °C were estimated using differential scanning calorimetry on STA 449 Jupiter, Hong Kong, Netzsch and dilatometry on NETZSCH DIL 402SU, Selb, Germany.

Two sets of the samples were supertransus solution treated (ST) at 930 °C for 30 min. One set was water quenched (WQ), and another was furnace cooled (FC), which corresponded to cooling rates of ~100 and ~0.1 °C/s, respectively.

Samples for EBSD investigations were prepared on a twin jet electropolisher (Fischione, Export, PA, USA) in 90% CH_3_COOH + 10% HClO_4_ electrolyte at a voltage of 22–25 V. Scanning electron microscopy was performed at 20 kV on TESCAN Mira3 LMU, Brno, Czech Republic, equipped with a Nordlys Nano EBSD. The scanning step was 0.4 or 0.05 µm, depending on the recognition quality of the Kikuchi line patterns. The error in determining the orientation of the crystal lattice was no more than ±1° (±0.5° on average).

The reconstruction of the high-temperature parent β-phase was carried out using the AZtech Crystals software according to the methods described in [27,28,41]. Pole figures (PFs) were employed for texture analysis of the α- and β-phases.

Electron backscatter diffraction (EBSD) was employed to evaluate the characteristics of α″(α)-crystallites, specifically the average size and lengthening (aspect ratio) of crystallites. The Ferret diameter of the region delineated by boundaries with misorientation angles greater than 10° was used as the average crystallite size. The regions with an area smaller than 1 μm^2^ were not taken into consideration.

TEM examination of thin foils was conducted using a Talos F200X microscope (USA) at 200 kV equipped with x-Act 6 and Ultim detectors for energy-dispersive X-ray spectroscopy (EDS).

X-ray diffraction (XRD) analysis was conducted using a D8 ADVANCE, Germany, diffractometer with Kα Cu radiation.

The misorientation between α-phase crystallites (α^i^) was calculated using the rotation matrix *M* obtained from the formula *α*^1^
*= Mα*^2^. The matrices α^1^ and α^2^ were calculated from the experimentally determined Euler angles using EBSD [41,42]. To calculate twin misorientations, the resulting rotation matrix R was compared with all variants of known twinning matrices according to the formula:H_ij_ = C_j_R^ − 1^C_i_T(1)
where T is the matrix T_1_, which describes the variant of the twin systems {$10\overset{-}{1}2$}, {$10\overset{-}{1}1$}, {$11\overset{-}{2}2$} и {$11\overset{-}{2}1$}, according to [41,43,44]; C_i_(C_j_) matrices describe the relationship of the initial basis with its crystallographically equivalent positions in the lattice (symmetry matrices). Only the type of phase lattice determines the number and type of C_i_ matrices, i.e., matrix representations of the rotational symmetry operator for the HCP lattice of Zr.

For each matrix H, the values of the Θ angles were calculated taking into account the full range of its symmetrical variants using the formula:Θ = arccos{(*h*_11_ + *h*_22_ + *h*_33_ − 1)/2}(2)
where *h_ij_* are the elements of the matrix H. Only one variant with the smallest value was chosen from the obtained sets of misorientation Θ-angles related to the matrix H. The misorientation between crystallites with Θ ≤ 2° was attributed to a certain type of twinning.

To determine the deviation from BOR, the possible misorientations between α and β phases for the β→α→β transformation path were calculated by a matrix method similar to that described in [45,46].

## 3. Results and Discussion

The analysis of the Zr-Nb equilibrium phase diagram [47] shows that during cooling from the β-region at an infinitely slow rate, the following sequence takes place in the Zr-2.5Nb alloy:(1)Precipitation of α-phase HCP crystallites with the gradual replacement of Nb with Zr;(2)Invariant transformation according to the eutectoid reaction of the remaining β-solid solution into α- and β_II_-phases (the latter is significantly enriched with Nb compared with the parent β_o_-phase); an increment of the β_II_-phase fraction is accompanied by an increase in Nb content up to almost 100 at. %.

The microstructure of the supertransus solution-treated sample (Figure 1a,b) is characterized by large parent polyhedral β-grains (20–200 μm) for both cooling rates. ST + FC from single phase β-region led to the formation of a typical tweed microstructure [48]: a Widmanstätten structure with nonparallel α-platelets with an average size of 4.3 μm, aspect ratio 3:1, where α-platelets are formed in many different planes within one parent β-grain (Figure 1c and Figure 2a). Note that the dimensions of all α-plates are considerably smaller than the size of the β-grains, meaning that the nucleation sites of the α-plates within the β-grains were the boundaries of the α-plates that formed earlier. This observation suggests that the FC rate is significantly higher compared to the cooling rate with which the phase diagrams are obtained. It appears that the microstructure was formed through a diffusionless transformation under near-isothermal conditions at a temperature notably below the β-transus.

After ST + WQ, the lenticular microstructure is formed with the separated α′-platelets (up to 100 μm) length comparable to the size of the parent β-grains and regions of significantly refined α′-platelets (3 μm, aspect ratio 4:1) (Figure 1d and Figure 3a). At higher magnifications (Figure 3a), it can be clearly seen that large α′-platelets can be considered as a colony of alternating α′-lamellae of the same orientation, morphologically resembling annealing twins. Crystallographic analysis revealed that some of these platelets are characterized by a misorientation angle of ~60° around the a_α_-axis and can be classified as {$10\overset{-}{1}1$}-compression twins [49]. The rest of the plates in the same colony are characterized by ~85° misorientation around the a_α_-axis with its neighbors and can be classified as {$10\overset{-}{1}2$}-tension twins [49]. Thus, α′-colonies consisting of relatively large alternating α′-platelets with twin misorientations were formed during cooling between the β-transus and the temperature of the invariant transformation. The simultaneous formation of twins of various types is apparently due to a compensation of elastic stress during the β→α′ phase transformation.

The regions with refined α′-platelets are characterized by greater morphological inhomogeneity (fineness, sharpness of crystallites) compared to the structure of the ST + FC sample (Figure 1c,d, Figure 2a and Figure 3a), which, obviously, implied more martensite-type structure.

As can be seen from Figure 1a,b, the reconstructed parent-β grains have a polyhedral structure with well-defined grain boundaries. One can mention a difference in the morphology of grain boundaries of WQ and FC samples. Grain boundaries of the WQ sample are characterized by an irregular morphology, which can be associated with the α-colonies growth mechanism. Namely, if two neighboring parent β-grains share common crystallographic planes, then a grain boundary α-variant is formed, sharing BOR with both β-grains. Thus, one α-variant can grow within two adjacent β-grains and, therefore, result in an irregular shape of the reconstructed β-grain boundaries [50]. In contrast, grain boundaries between two adjacent β-grains form flat facets in the WQ sample due to successful relaxation processes during fast cooling. The presence of low-angle boundaries within the parent β-grains on the reconstructed maps is apparently due to the fact that phase transformations take place with a slight deviation from the Burgers OR.

In general, the obtained results of the microstructure characteristics align with the findings from [14,25,26]. Some discrepancies, such as differences in alloy chemical compositions, may be attributed to the variation in experimental conditions [14,26], such as the superheating temperature during ST [25] and annealing in the capsule [14,25]. These factors influence both the actual cooling rates and the critical cooling rates, which affect the kinetics of the transformation processes and the size and morphology of the α(α′)-phase crystallites.

The peaks at the angles near 60 and 90° appear at the misorientation angle distribution (MAD) of the α(α′)-phase after both FC and WQ (Figure 1, Figure 2c and Figure 3c). Minor α(α′)/α(α′)-boundaries with misorientation angles of about 10° are also observed, which is consistent with the results of [25]. Such discrete α(α′)/α(α′)-MAD spectra point to the strict crystallographic limitations during the β→α(α′)-phase transformation, which, in our opinion, can only be realized by shear and shuffle mechanisms [50,51], i.e., with a strictly directed rearrangement of atoms during the transformation of the crystal lattice.

The phase maps demonstrate that the β-phase is enough for a misorientation analysis (Figure 2b, Figure 3b and Figure 4a,b). A quantitative EBSD analysis revealed a significantly higher fraction of β-phase in the FC (~3%) sample compared to the WQ (~0.2%) sample. The volume fraction of the phases is consistent with the results of [25]. A significantly higher fraction of the β phase in the FC microstructure confirms the nonequilibrium nature of the transformations. It is likely that the β-phase during FC changed its chemical composition closer to the eutectoid one. The major fraction of the β-phase was observed at the boundaries of the α-colonies and the parent β-grains boundaries in the FC sample. A minor fraction of the β-phase is located within the α-colonies and is attached to the low-angle boundaries (Figure 2a,b). In the case of WQ, there are no fine precipitates of the β-phase within α′-crystallites, and they are observed only at high-angle boundaries (Figure 3a,b).

The analysis of deviation from the BOR spectra for the FC sample showed that α/β boundaries generally follow BOR, and the scattering does not exceed 4° (Figure 4c). However, some peaks systematically appear at specific angles close to 29°, 35°, and 43° (Figure 2c). This indicates the presence of at least three additional specific crystallographic orientations of the β-phase formed in the parent β-grain. For the WQ sample, the fraction of non-Burgers β-phase orientations is significantly higher (Figure 3d and Figure 4d).

The observed patterns, both in the spectra of intercrystalline boundaries and in the spectra of deviation from the BOR, are relevant for both large EBSD investigation areas, including a significant number of initial β-grains, and for areas corresponding to individual β-grains (highlighted in Figure 4a,b). Pole figures (Figure 4e,g) for the selected area in the FC sample (Figure 4a) reveal a single intense β-phase orientation and 12 major α-phase orientations of varying intensities (six reflections on the {0001} IPF) that are BOR-connected to the β-phase orientation. Consequently, the observed β-phase orientation coincides with the initial orientation of the β-grain. In the WQ sample, BOR is fulfilled for the 12 primary α-phase orientations (Figure 4h) and a single β-orientation (i.e., they are formed in accordance with BOR in a single parent β-grain). Simultaneously, numerous β-orientations are observed in this region (Figure 4f), for only one of which (and not the most intense one) BOR is fulfilled with α′-phase crystallites. Thus, the majority of β-phase precipitates are not retained and formed as a result of more complex processes.

EBSD data for FC and WQ samples at higher magnification (Figure 2c and Figure 3c) revealed that the majority of α(α′)/α(α′)-boundaries are also characterized by misorientation angles close to 10°, 60° and 90° corresponding to the orientations formed according to BOR. Additional weak peaks are observed on α′/α′ MADs for the WQ sample at the angles in the ranges of 28–31° and 70–76° (Figure 3c).

It should be noted that the 60° and 90° misorientations are quite close to the {$10\overset{-}{1}1$} and {$10\overset{-}{1}2$} twins, respectively. Thus, one can conclude that both FC and WQ microstructures can be characterized as consisting predominantly of phase transformation twins [49].

All β-precipitates are located at the α(α′)/α(α′)-boundaries for both cooling rates. β-precipitates are noticeably larger in the FC sample compared to the WQ sample (Figure 2a,b). The major fraction of β-precipitates is typical of 10° α/α-misorientation angles; the minor β-fraction corresponds to the 60°-boundaries and the least to 90°-boundaries. The major fraction of β-precipitates within one parent β-grain coincides with the orientation of the parent one (BOR is fulfilled with the neighboring α-platelets). However, a small number of β-precipitates observed at α/α-boundaries with a misorientation of 10° are characterized by orientations significantly different from the initial β-grain. These precipitates correspond to the peaks 29°, 35° and 43° in deviation from BOR spectra.

The small size of β-precipitates in the WQ samples does not allow us to link their orientation with the crystallography of α′/α′ grain boundaries. While the strongest peaks are observed at angles of 29°, 35° and 43°, the perfect BOR peak (~0°) turns out to be significantly weaker. The observed peaks in the spectra of deviation from BOR are more scattered compared to the FC sample.

A theoretical crystallographic calculation of the possible α′/β-misorientations angles according to BOR showed that the observed deviations from BOR at angles close to 29, 35, and 43° correspond to crystallographic equivalents of BOR in the course of β→α→β-transformation. Alternatively, in other words, the appearance of such β-orientations is possible if these β-precipitates are formed in the course of the β_o_→α(α′)→β_II_-transformation path, where each transformation stage follows BOR. However, the BOR pathway for the α(α′)→β_II_ transformation stage is not the same as the BOR for the β_o_→α(α′) transformation stage.

XRD (Figure 5) generally corroborates the results of EBSD. In the case of FC, the sample predominantly consists of the α-phase with lattice parameters a = 0.3227 nm and c = 0.5139 nm, along with a significant fraction of β-phase. In the case of water quenching, the sample is characterized by α′-phase with lattice parameters a = 0.3228 nm and c = 0.5154 nm; the β-phase is not detected, likely due to its small fraction and size of precipitates. The Nb content of about 20 at. % in the β-phase estimated from the lattice spacing determined by XRD was close to the eutectoid composition for the FC sample.

The TEM microstructure of the FC sample is characterized by α-phase colonies consisting of alternating platelets (Figure 6), which is in good agreement with EBSD data. Thin β-phase precipitates are clearly resolved between neighboring α-platelets. Relatively small deformation twins are observed within the α″-lamellae (Figure 6b).

EDS analysis of the FC sample (Figure 6c,d) showed that the α-phase contains 99.5 at. % Zr and 0.5 at. % Nb and the β-phase contain 87 at. % Zr and 13 at. % Nb. Note that Zr is the main element in the β-phase, and its composition only approaches equilibrium, corresponding to the maximum solubility of Zr in the β-phase before the invariant transformation (~80 at. %). However, the phase diagram implies that during slow cooling, the β-phase contains up to 97 at. % Nb at room temperature.

Analysis of the Fast Fourier Transform (FFT) from the high-resolution transmission electron microscopy (HRTEM) image revealed that α-regions represent two orientations misoriented relative to each other by 30°, which correspond to the stacking faults in HCP (Figure 6e,f). The β-regions contain nanosized precipitates of the secondary α-phase (Figure 6e,g) along with large first-order α-platelets characterized by a single crystallographic orientation. Moreover, TEM demonstrates β-phase orientations that deviate from BOR at angles of 29° (Figure 6e), 35° and 43° with the neighboring α-phase as well as the predominant crystallographic orientation of the β-phase, which follows BOR with the neighboring α″-platelets.

The TEM microstructure of the WQ sample is characterized by comparably large alternating α′-platelets (Figure 7a,e), which were recognized as transformation twins according to EBSD analysis. In addition, parallel α′-strands with a thickness of about 50 nm are observed within these first-order α′-platelets (Figure 7e,f). These α′-strands can be identified as deformation twins due to typical block internal structure (Figure 7f) and diffuse scattering of α-phase reflections in FFT images (Figure 7h). Homogeneous regions of the α-phase marked 2 and 3 in Figure 7f contain a high number of stacking faults and can be considered as two HCP orientations misoriented relative to each other by 30° (Figure 7g,i). EDS analysis (Figure 7c,d) showed an almost homogeneous distribution of Zr and Nb in the microstructure of the WQ sample.

The investigation of microstructures obtained at substantially different cooling rates from the β-region at various magnification scales reveals the following phenomenology of phase transformations in the Zr-2.5%Nb alloy. As the temperature falls below the β-transus during FC but remains above the invariant transformation temperature, a martensitic (displacive) transformation β→α′ occurs in accordance with BOR. The chemical composition of the α′-phase at the time of transformation is similar to that of the alloy. During the transformation process, the β_II_ phase is formed on crystallographically determined (special) intercrystalline boundaries α′/α′ in accordance with BOR, with the most probable variant of its orientation returning to the orientation of the parent β_o_ grain [36]. Further cooling down to the invariant transformation temperature results in the enrichment of the β-precipitations with Nb to approximately 20 at. % at eutectoid point. During cooling below the invariant transformation temperature, the β_II_-phase becomes metastable, while diffusion processes at low temperatures are significantly slowed down. Consequently, decomposition of the β_II_ phase occurs with the precipitation of refined α-phase in accordance with BOR.

The martensitic β→α″-transformation takes place at a lower temperature during WQ, closer to the invariant transformation temperature, when diffusion processes are almost inhibited. During the diffusionless β→α′ transformation, a tiny fraction of the β_II_ phase is formed in accordance with one of the potential variants of BOR.

The proposed phenomenology of the phase transformation upon cooling of the Zr-2.5Nb alloy is in good agreement with [52], where the temperature of the martensitic transformation of the Zr-2.5Nb alloy estimated by XRD and thermal analysis was shown to significantly exceed the temperature of the invariant transformation.

## 4. Conclusions

This study investigates the morphological characteristics of (α(α″) + β)-microstructures in the Zr-2.5Nb alloy and the crystallographic relationships between α(α′)- and β-phases due to β→α(α′)-transformation occurring during furnace cooling (FC) and water quenching (WQ). Notable dimensional and morphological differences were observed in the precipitates of the α-, α″-, and β-phases formed under varying cooling rates. When comparing FC to WQ, the microstructure exhibits improved grain (morphological) homogeneity (uniform α crystallite sizes) and pronounced chemical heterogeneity (resulting from the presence of large β precipitates in the structure). Analogous crystallographic regularities of phase transformations for both cooling rates imply their predominantly displacive nature. The misorientation angle distribution spectra of intercrystallite boundaries for both cooling rates align with phase transformations in accordance with BOR.

The precipitates of the secondary Nb-enriched β_II_-phase crystallographically different from the high-temperature parent β-phase were characterized by peaks of deviation from the BOR spectra near angles close to 0, 29, 35, and 43° for both FC and WQ. Theoretical calculations of α/β misorientation, based on BOR along the β→α→β-transformation route, confirmed the experimental spectra. The intensity of the angular distribution is affected by the cooling rate: the most significant deviation from BOR at angles near 29°, 35°, and 43° is more characteristic of the WQ samples compared to FC samples. Despite the substantial differences in the α-, α″-, and β-morphologies of Zr-2.5Nb after FC and WQ, a clear crystallographic inheritance during the β_o_→α′(α)→β_II_-transformation suggests similar mechanisms of lattice rearrangement. Moreover, this finding highlights the crucial role of shear and shuffle in phase transformations during the cooling of the Zr-2.5%Nb alloy from the single-phase β-region, independent of the cooling rate.

The acquired data on the crystallographic relationship between the α(α″)- and β-phases during phase transformations at any cooling rate should enable the design of materials and products with optimal operational performance for critical applications. The microstructure analysis supports the recommendation of FC for structural materials in nuclear reactors, where resistance to radiation creep is of vital importance and WQ for materials and products requiring high corrosion resistance in the chemical and nuclear industries.

## Figures and Tables

**Figure 1 materials-16-03758-f001:**
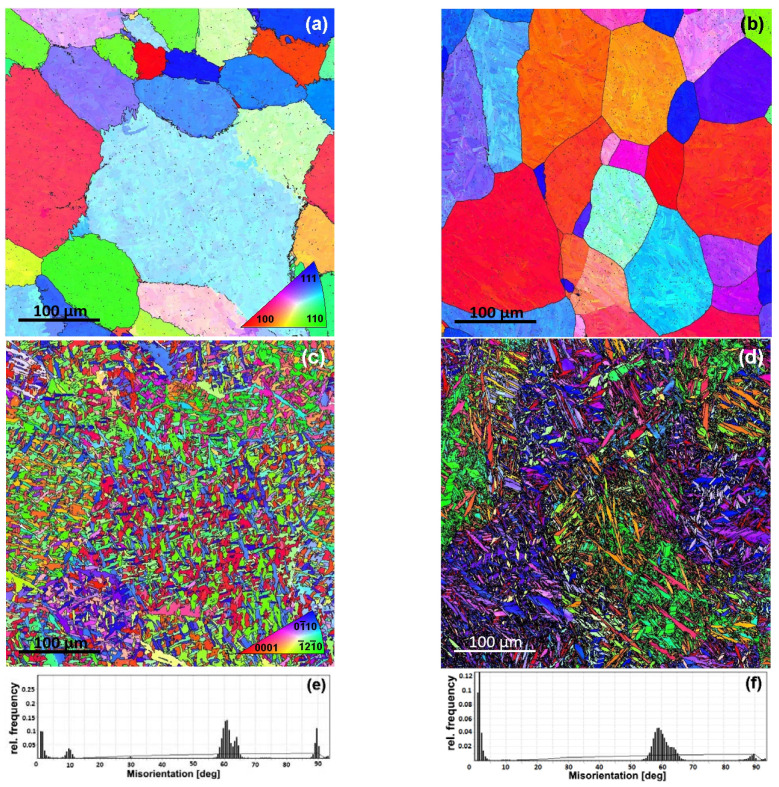
Parent β-grains reconstruction of super solution treated Zr-2.5Nb alloy: (**a**) FC, (**b**) WQ samples; the IPF colored maps for: (**c**) FC, (**d**) WQ samples; the misorientation angle distribution in the α-phase for: (**e**) FC, (**f**) WQ samples.

**Figure 2 materials-16-03758-f002:**
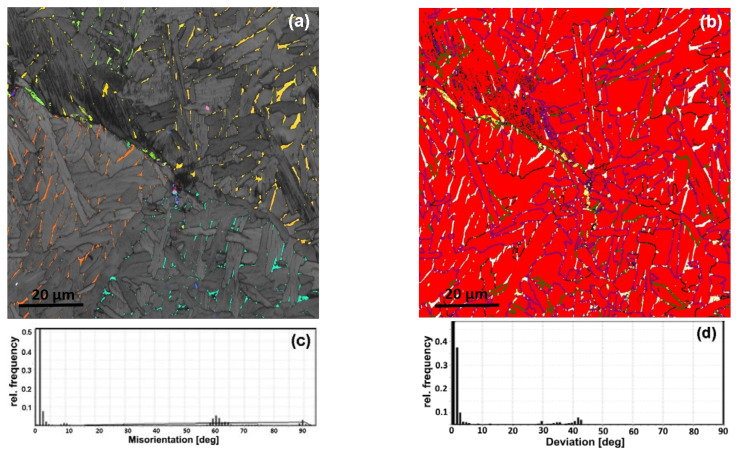
Band contrast with PF colored β-phase for super solution treated and FC Zr-2.5Nb (**a**); (**b**) phase maps with α-phase—red (black—boundaries between α-crystallites misoriented at 2–15°, violet—boundaries between α-crystallites misoriented 15–80°, green—boundaries between α-crystallites misoriented 80–95°), β-phase—white (yellow color characterizes the α/β-boundaries—deviated from BOR at angles over 10°); (**c**) MAD in the α-phase and deviation from BOR spectrum (**d**).

**Figure 3 materials-16-03758-f003:**
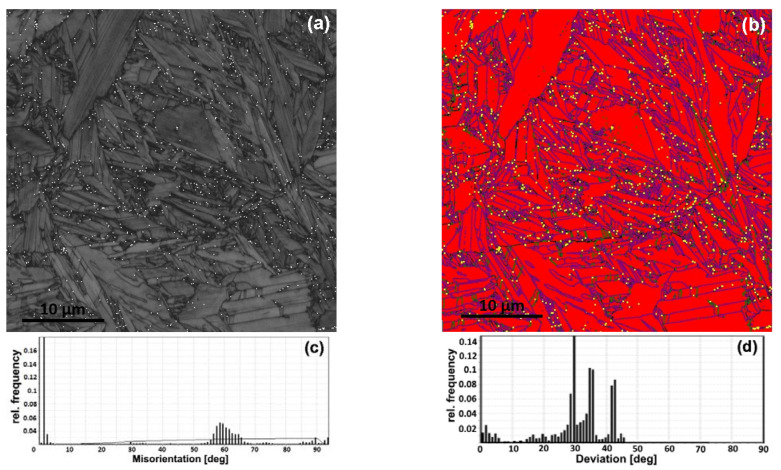
Band contrast with β-phase highlighted white for super solution treated and WQ Zr-2.5Nb (**a**); (**b**) phase map with α-phase—red (black—boundaries between α-crystallites misoriented at 2–15°, violet—boundaries between α-crystallites misoriented 15–80°, green—boundaries between α-crystallites misoriented 80–95°) and β-phase—white (yellow color characterizes the α/β-boundaries deviated from BOR at angles over 10°); (**c**) MAD in the α-phase; (**d**) spectra of deviation from BOR.

**Figure 4 materials-16-03758-f004:**
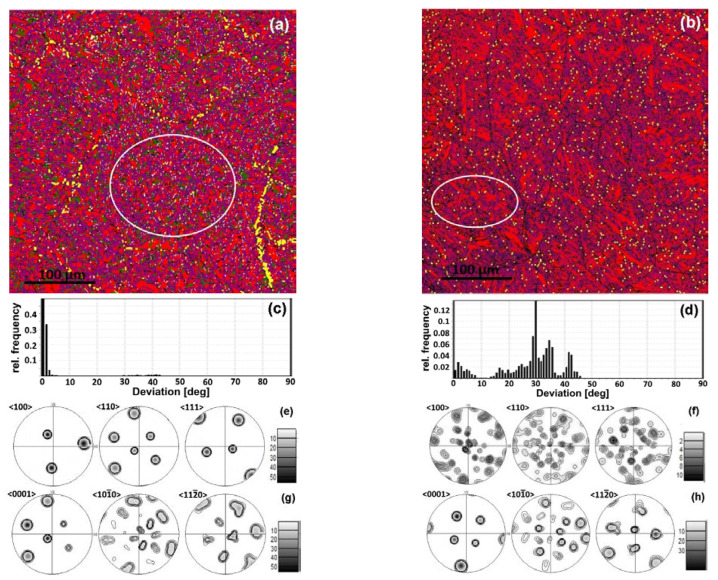
Phase maps for super solution treated Zr-2.5Nb alloy: (**a**) FC, (**b**) WQ samples (α-phase—red; β-phase—blue; yellow color characterizes the α/β-boundaries deviated from BOR); spectra of deviation from BOR for: (**c**) FC, (**d**) WQ samples; (**e**,**f**)—corresponding β-phase PFs; (**g**,**h**)—corresponding α-phase PFs; the regions taken for misorientation analysis for single β-grain are marked with ovals.

**Figure 5 materials-16-03758-f005:**
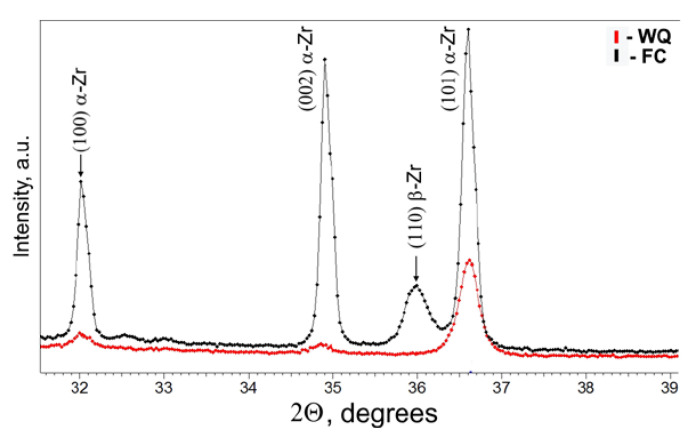
XRD patterns for Zr-2.5Nb alloy after WQ and FC.

**Figure 6 materials-16-03758-f006:**
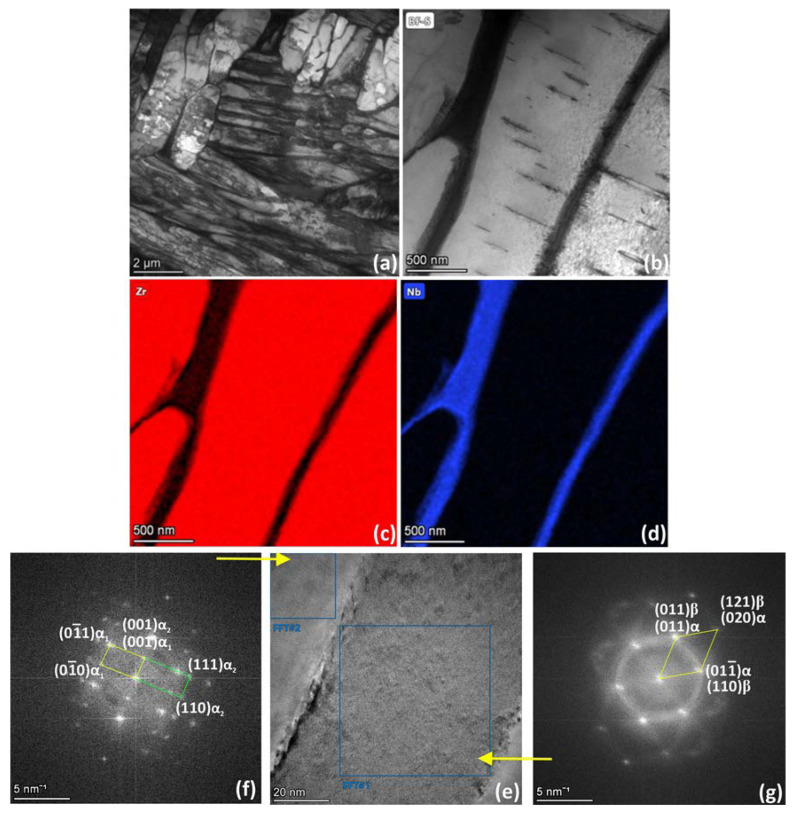
TEM results for super solution treated and FC Zr-2.5Nb: (**a**,**b**) bright field images, (**c**,**d**) EDS elemental maps (Zr, Nb); (**e**–**g**) high-resolution transmission electron microscopy (HRTEM) image and corresponding FFTs: (**f**) zone axis (100)α_1_, ($1\overset{-}{1}0$)α_2_; (**g**) zone axis (100)α, ($1\overset{-}{1}1$)β.

**Figure 7 materials-16-03758-f007:**
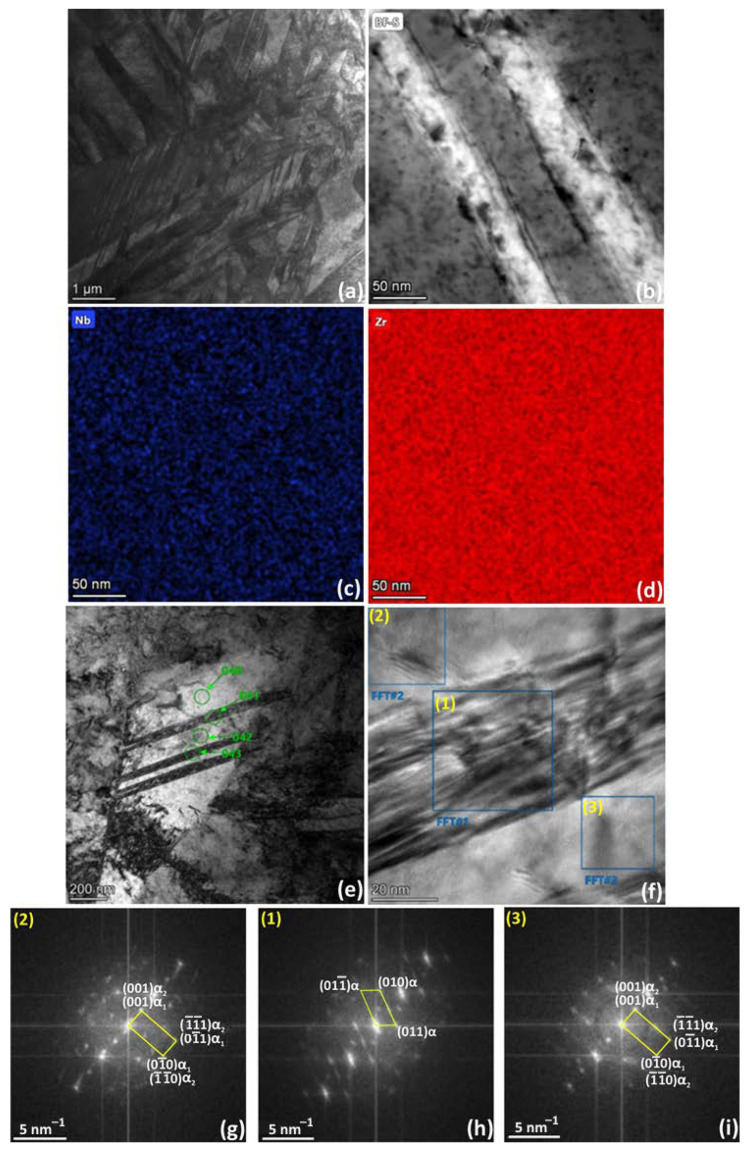
TEM results for super solution treated and WQ Zr-2.5Nb: (**a**,**b**,**e**) bright field image, (**c**,**d**) EDS elemental maps (Zr, Nb); (**f**–**i**) HRTEM image and corresponding FFTs: (**g**) zone axis ($\overset{-}{1}00$)α_1_, ($\overset{-}{1}10$)α_2_; (**h**) zone axis ($\overset{-}{1}00$)α; (**i**) zone axis ($\overset{-}{1}00$)α_1_, ($\overset{-}{1}10$)α_2_.

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
