# Peer review of "The Effect of Cooling Rate on Crystallographic Features of Phase Transformations in Zr-2.5Nb"

_materials, 2023, doi:10.3390/ma16103758_

Round 1
Reviewer 1 Report
The authors investigated the crytallographic features of phase transformation of Zr-2.5Nb at different cooling rates. Differences and similarities observed through EBSD and TEM images were described, but quantitative analysis considering relevance to application fields was not sufficiently provided.
1. What is the new finding of the research, and what does it mean to the field of interest?
2. How can the results of the research be used? For example, can you suggest what kind of physical properties are required, and which cooling rate is better for the properties? Or can you explain the importance and the effect of the ‘similar mechanisms of lattice rearrangement’ on the field?
3. What does the 'significant dimensional and morphological difference' in conclusion means? They are not specific nor quantitative.
4. What is the meaning of the precise crystallographic inheritance, and how does it support the ‘similar mechanisms of lattice rearrangement’? It seems that the WQ sample mostly followed BOR.
5. How can you conclude that the shear and shuffle play a significant role in the phase transformation?
Author Response
|
Reviewer â„–1 |
|
|
We gratefully thank the reviewer for valuable recommendations, according to which we made a major revision of the manuscript. In addition, we apologize for the delay with the revision, which was caused by the necessity to conduct additional experimental research and to make some essential corrections related to microstructure analysis. The experimental observations allowed us to expand the phenomenology description of phase transformation during cooling. Moreover, we made comprehensive revision of Introduction in order to better connect the study's objectives with existing theory and research. We also included more current references to highlight the relevance of the research. All edits are marked in the manuscript by blue highlighting. |
|
|
Comment |
Reply |
|
The authors investigated the crytallographic features of phase transformation of Zr-2.5Nb at different cooling rates. Differences and similarities observed through EBSD and TEM images were described, but quantitative analysis considering relevance to application fields was not sufficiently provided. |
The manuscript is supplemented with quantitative characteristics of microstructures obtained at different cooling rates. In addition, we added recommendations for cooling rates relative to application fields to Conclusions based on metallographic analysis. |
|
1. What is the new finding of the research, and what does it mean to the field of interest?
|
The research establish the crystallographic regularities of the microstructure evolution in course of α↔β phase transformation during cooling. Based on the experimental data analysis, the phenomenology of phase transformations in the Zr-2.5%Nb alloy was formulated (please find it in the last paragraphs of Results and Discussions). In accordance these edits, the purpose of the work and the Conclusions have been slightly modified. |
|
2. How can the results of the research be used? For example, can you suggest what kind of physical properties are required, and which cooling rate is better for the properties? Or can you explain the importance and the effect of the ‘similar mechanisms of lattice rearrangement’ on the field?
|
The results of the work allows for tailoring the microstructure of Zr alloy based alloys and products with optimal operating performance for their various applications.
We added recommendations for the use of various cooling rates to obtain products with a certain microstructure for critical applications in the Conclusions.
|
|
What does the 'significant dimensional and morphological difference' in conclusion means? They are not specific nor quantitative.
|
We added the estimation of quantitative characteristics of microstructures obtained via different cooling rates. |
|
What is the meaning of the precise crystallographic inheritance, and how does it support the ‘similar mechanisms of lattice rearrangement’? It seems that the WQ sample mostly followed BOR.
|
The mechanisms of texture inheritance during bcc-hcp-bcc phase transformations are comprehensively analyzed in Tomida et. al . 2021 (corresponding reference [36] was added in revised version). The manuscript was supplemented with some explanations (final paragraphs of Results and Discussions). |
|
How can you conclude that the shear and shuffle play a significant role in the phase transformation?
|
Only the shear and shuffle lattice rearrangement provide a strictly directed displacement of atoms during the transformation of the crystal lattice, thus, ensure the fulfillment of BOR. Some explanations are included in the text of the article. |

Reviewer 2 Report
overall, the paper is well written, and direct to the point.
General comments:
Abstract:
The abstract of the manuscript does not accurately convey the main findings of the study, and as such, I strongly recommend that the authors revise it to improve its clarity and ensure that it accurately represents the results and conclusions of their research.
Introduction part:
One area that could benefit from further development is the theoretical framework used in the paper. I would suggest revisiting the literature review to better connect the study's objectives with existing theory and research.
1. Overall, I found the paper to be well-researched and thoughtfully written. However, I would suggest revisiting the introduction to provide clearer context for the research question and motivation for the study." Additional information, sources for utilizing nuclear materials , and citations would enhance the credibility and clarity of the introduction which may discusses the using of a new ATF materials within the nuclear fuel. The authors may benefit from consulting the literature on this topic to further develop and support their arguments.
a. https://doi.org/10.1016/j.nucengdes.2021.111486
b. https://doi.org/10.3390/en15103772
2. Provide a brief discussion about different types of and fission nuclear power plant (LWR, BWR, HTR)
a. https://www.sciencedirect.com/topics/engineering/light-water-reactors
3. Figures 1 and 2: Could you provide more details about the content of figure one?
4. I noticed a few instances of unclear language and grammatical errors throughout the paper. Addressing these issues would improve the readability and overall impact of the study.
5. The reviewer would recommend the authors to provide a table of abbreviation in the manuscript
Conclusion
conclusion could be strengthened by providing more explicit recommendations for future research and practice. This would help ensure that the study's contributions are fully realized and applied in relevant contexts.
Author Response
|
Reviewer â„–2 |
|
|
We gratefully thank the reviewer for valuable recommendations, according to which we made a major revision of the manuscript. In addition, we apologize for the delay with the revision, which was caused by the necessity to conduct additional experimental research and to make some essential corrections related to microstructure analysis. The experimental observations allowed us to expand the phenomenology description of phase transformation during cooling. Moreover, we made comprehensive revision of Introduction in order to better connect the study's objectives with existing theory and research. We also included more current references to highlight the relevance of the research. All edits are marked in the manuscript by blue highlighting. |
|
|
Comment |
Reply |
|
Abstract: The abstract of the manuscript does not accurately convey the main findings of the study, and as such, I strongly recommend that the authors revise it to improve its clarity and ensure that it accurately represents the results and conclusions of their research. |
The results of the study were clarified and expanded in the Abstract within the allowed limit of 200 words. |
|
Introduction part: One area that could benefit from further development is the theoretical framework used in the paper. I would suggest revisiting the literature review to better connect the study's objectives with existing theory and research. |
The Introduction has been comprehensively revised to include the results of recent research on the subject. Introduction and appropriate References have been significantly reworked in order to connect the study's objectives with existing theory and research.
|
|
1. Overall, I found the paper to be well-researched and thoughtfully written. However, I would suggest revisiting the introduction to provide clearer context for the research question and motivation for the study." Additional information, sources for utilizing nuclear materials , and citations would enhance the credibility and clarity of the introduction which may discusses the using of a new ATF materials within the nuclear fuel. The authors may benefit from consulting the literature on this topic to further develop and support their arguments.
a. https://doi.org/10.1016/j.nucengdes.2021.111486 b. https://doi.org/10.3390/en15103772 |
We made a thorough revision of the Introduction. Since the article is devoted to the crystallographic aspects of microstructure evolution, the emphasis shifted towards the describing the main problems (radiation growth, corrosion resistance, hydrogenation) associated with the microstructure and texture formation of Zr alloys used in nuclear reactors,.
We thank the Reviewer for the recommended references. They helped to understand the discussed problems in more details and were partially used in the References list.
|
|
2. Provide a brief discussion about different types of and fission nuclear power plant (LWR, BWR, HTR) a. https://www.sciencedirect.com/topics/engineering/light-water-reactors |
|
|
3. Figures 1 and 2: Could you provide more details about the content of figure one? |
We added quantitative estimates and explanations to the Figures 1 and 2 and their relationships. |
|
4. I noticed a few instances of unclear language and grammatical errors throughout the paper. Addressing these issues would improve the readability and overall impact of the study. |
We reworked the manuscript to eliminate these errors. |
|
5. The reviewer would recommend the authors to provide a table of abbreviation in the manuscript. |
We use rather common abbreviations and provide abbreviation expansions in the manuscript. |
|
Conclusion conclusion could be strengthened by providing more explicit recommendations for future research and practice. This would help ensure that the study's contributions are fully realized and applied in relevant contexts. |
The conclusions have been substantially revised. The description of possible applications for design of Zr alloys microstructures and components made of Zr alloys have been added as well. |

Reviewer 3 Report
Upon reviewing your manuscript intitled: The effect of cooling rate on crystallographic features of phase transformations in Zr-2.5Nb. I find your work interesting, but I do not believe it can be published in its current form. Therefore, some revisions must be done so that it may be published in this journal. I have the following comments and recommendations.
- The introduction of relevant background and research progress was not comprehensive enough.
- There are many examples of related research in the introduction, which lacks the author's own understanding and elaboration. Please cite the literature reasonably and add some of your own views.
- An X-ray diffraction phase study should be shown to confirm the results presented by TEM.
- Use HRTEM and determine the crystallographic planes of the material, check with the XRD ray diffraction analysis that should be included.
- Based on the results obtained, how or where can these materials be applied?
- It is appreciable that the authors have studied: The effect of cooling rate on crystallographic features of phase transformations in Zr-2.5Nb, but the results are descriptive and some hypothetical, needing to be discussed more strongly.
- It is recommended to include more current references to highlight the relevance of the research.
Author Response
|
Reviewer â„–3 |
|
|
We gratefully thank the reviewer for valuable recommendations, according to which we made a major revision of the manuscript. In addition, we apologize for the delay with the revision, which was caused by the necessity to conduct additional experimental research and to make some essential corrections related to microstructure analysis. The experimental observations allowed us to expand the phenomenology description of phase transformation during cooling. Moreover, we made comprehensive revision of Introduction in order to better connect the study's objectives with existing theory and research. We also included more current references to highlight the relevance of the research. All edits are marked in the manuscript by blue highlighting. |
|
|
Comment |
Reply |
|
- The introduction of relevant background and research progress was not comprehensive enough.
|
The Introduction has been thoroughly revised according to Reviewer comments.
Appropriate revision of References have been made.
|
|
- There are many examples of related research in the introduction, which lacks the author's own understanding and elaboration. Please cite the literature reasonably and add some of your own views.
|
|
|
- An X-ray diffraction phase study should be shown to confirm the results presented by TEM.
|
The XRD results are provided in Fig. 6 of the revised version and their correlation with the TEM results is discussed. |
|
- Use HRTEM and determine the crystallographic planes of the material, check with the XRD ray diffraction analysis that should be included.
|
The TEM results have been reworked. The Fast Fourier Transform interpretation is added and discussed along with XRD data. |
|
- Based on the results obtained, how or where can these materials be applied?
|
The description of possible applications for design of Zr alloys microstructures and components made of Zr alloys have been added in Conclusion. |
|
- It is appreciable that the authors have studied: The effect of cooling rate on crystallographic features of phase transformations in Zr-2.5Nb, but the results are descriptive and some hypothetical, needing to be discussed more strongly.
|
Please find the corresponding discussion in the final paragraphs of the revised version of the manuscript. |
|
- It is recommended to include more current references to highlight the relevance of the research. |
The References has been substantially changed in favor of relevant literature. |

Round 2
Reviewer 1 Report
I think this version of draft can be accepted for publication.
Reviewer 3 Report
This version may be considered for publication.